# Impact of Chemical Properties of Human Respiratory Droplets and Aerosol Particles on Airborne Viruses’ Viability and Indoor Transmission

**DOI:** 10.3390/v14071497

**Published:** 2022-07-08

**Authors:** Ajit Ahlawat, Sumit Kumar Mishra, Hartmut Herrmann, Pradhi Rajeev, Tarun Gupta, Vikas Goel, Yele Sun, Alfred Wiedensohler

**Affiliations:** 1Leibniz Institute for Tropospheric Research, 04318 Leipzig, Germany; herrmann@tropos.de (H.H.); alfred.wiedensohler@tropos.de (A.W.); 2CSIR-National Physical Laboratory, New Delhi 110012, India; sumitkumarm@gmail.com; 3Department of Civil Engineering, Indian Institute of Technology (IIT), Kanpur 208016, India; pradhi.iitk@gmail.com (P.R.); tarun@iitk.ac.in (T.G.); 4School of Interdisciplinary Research, Indian Institute of Technology (IIT), Delhi 110016, India; vikasgoel1002@gmail.com; 5LAPC, Institute of Atmospheric Physics, Chinese Academy of Sciences, Beijing 100017, China; sunyele@mail.iap.ac.cn

**Keywords:** SARS-CoV-2, aerosol particles, droplets, chemical composition, pH, airborne

## Abstract

The airborne transmission of severe acute respiratory syndrome coronavirus 2 (SARS-CoV-2) has been identified as a potential pandemic challenge, especially in poorly ventilated indoor environments, such as certain hospitals, schools, public buildings, and transports. The impacts of meteorological parameters (temperature and humidity) and physical property (droplet size) on the airborne transmission of coronavirus in indoor settings have been previously investigated. However, the impacts of chemical properties of viral droplets and aerosol particles (i.e., chemical composition and acidity (pH)) on viability and indoor transmission of coronavirus remain largely unknown. Recent studies suggest high organic content (proteins) in viral droplets and aerosol particles supports prolonged survival of the virus by forming a glassy gel-type structure that restricts the virus inactivation process under low relative humidity (RH). In addition, the virus survival was found at neutral pH, and inactivation was observed to be best at low (<5) and high pH (>10) values (enveloped bacteriophage Phi6). Due to limited available information, this article illustrates an urgent need to research the impact of chemical properties of exhaled viral particles on virus viability. This will improve our fundamental understanding of indoor viral airborne transmission mechanisms.

## 1. Introduction

Respiratory droplets are quite complex in nature due to their varying physicochemical properties such as size, surface-to-volume ratio, and spatial heterogeneous composition. The virus which is embedded into droplets experiences dynamic and highly variable micro-environments. Droplets, when expelled from the human respiratory system, undergo rapid change with relative humidity (RH) from 100% to indoor room conditions [1,2,3]. These droplets evaporate rapidly, and the concentration of salt, protein, and inorganics increases by nearly one order of magnitude due to the loss of water content [1,2]. Liu et al. [4] assumed that the aqueous part of a respiratory droplet consisted of sodium chloride (NaCl) solution in their model for the purpose of simplification. However, for a droplet mainly consisting of NaCl and water, the evaporation rate is likely to increase at a faster rate, once a salt nucleus is formed and exceeds the critical size [5]. Therefore Liu et al. [4] reported a decrease in droplet diameter by a factor of 3. However, Vejerano and Marr [1] did not observe such an effect in their study. They demonstrate this behavior somewhat due to the presence of mucin and dipalmitoyl phosphatidylcholine (DPPC) which were experiencing phase changes. The evaporating expelled droplets, which do not fall to the ground due to their small size, are termed “aerosol particles” [6]. The airborne route of COVID-19 spread was acknowledged in recent studies around the world [7,8,9], and the airborne transmission of SARS-CoV-2 in indoor environments were subject to the influences of meteorological parameters such as temperature and RH [6,10,11,12]. While the physical properties of respiratory droplets and viral aerosol particles have dominated the discussion of airborne transmission [9]. However, there are only a few recent experimental and modeling studies on the role of chemical properties of aerosol particles in the virus spread and survival of the virus in indoor settings [13,14,15].

In this article, we provide information about respiratory droplets and aerosol particles’ chemical properties, including chemical composition and particle acidity (pH), to explain their impact on virus viability in aerosol particles. Although, this is a novel virus, and it would be difficult to find many studies directly linking to aerosol particle chemical properties and SARS-CoV-2. Therefore, we have assessed the enveloped viruses such as Influenza, Influenza A virus (IAV), Langat, the enveloped bacteriophage Phi6 virus, and SARS-CoV-1 and their association with chemical properties of droplets and aerosol particles. Here, we focus on reporting the impact of aerosol chemical properties on enveloped viruses only. In this article, we will describe the effect of chemical composition and pH on virus viability inside aerosol particles and droplets. We will further report the changes in droplet chemical properties due to indoor humidity, ultimately affecting virus viability and potential instrumentation available to measure expelled respiratory viral droplets and aerosol particles’ chemical properties.

## 2. Effect of Chemical Properties of Human Respiratory Droplets and Aerosol Particles in Virus Viability in Indoor Environments

### 2.1. Effect of Chemical Composition on Virus Viability of Enveloped Viruses

It is well recognized that physical properties, such as particle size of viral droplets, have an impact on virus viability. The droplets’ size is governed by their chemical composition. Therefore, the information on the impact of the chemical composition of respiratory droplets and aerosol particles on the viability of embedded viruses is very important [2,16,17]. Due to changes in the chemical properties of the aerosol particles and droplet nuclei, there may be changes in virus infectivity [16,18]. Typically, the generation of aerosol particles is considered from the lower (alveolar), middle (laryngeal), and upper (oral cavity) respiratory tract regions [19]. Therefore, the composition of respiratory droplets would depend on their tract regions [19]. However, the information on the chemical properties of viral droplets originating from various tract regions is not completely available. However, Bozic and Kanduc [20] reported the basic composition of respiratory droplets of various origins in tabular form. They mentioned that human respiratory droplets and aerosol particles majorly consist of organic and inorganic constituents (Figure 1). In the previous studies, major components reported as respiratory fluids are salt, proteins, and surfactants [21]. The main inorganic compound found in respiratory fluids is sodium chloride (NaCl) whereas the primary organic substances are lactate and glycoproteins [21,22]. Nicas et al. [23] reported that the aerosolized respiratory mucus consisted of sodium chloride (NaCl) (150 ± 20 mM, roughly 8.8 g/L) and protein (76 ± 18 g/L). It may be noted, however, that the sputum and secretions from other regions such as nasal, oropharyngeal, tracheobronchial, and bronchoalveolar have different compositions [24]. Furthermore, the viral droplet composition (chemical composition, pH) changes for different types of diseases [24]. The distribution of proteins of human mucosal fluids in different respiratory tract regions can be found in a table provided by Niazi et al. [2] based on previous studies [25,26]. In addition, Vejerano and Marr [1] reported salts, proteins, and surfactants as major constituents of human respiratory mucus. Therefore, when a measurement of the viral droplets and aerosol particles is carried out in environmental settings, the focus should be on the characteristics of aerosol particles (e.g., concentration of solutes, pH, size, etc.) which are exhaled from the human respiratory tract. There are some recent studies that have shown the effect of mucus on influenza survival in aerosol and droplets [27,28].

The transmission rates and environmental conditions are closely related when considering investigating respiratory disease [29]. The relative humidity and temperature of indoor air are the major environmental factors that can affect the viability of airborne respiratory viruses [30]. The physicochemical properties of viral droplets due to changes in indoor air humidity and temperature can vary due to the evaporation of the droplets [1], which can ultimately affect the viability of embedded viruses. As more water content evaporates from the droplets, there will be an increase in lower volatility solutes concentration, such as salts and organic compounds [29]. Previously, there are multiple hypotheses developed in order to explain the relationship between virus viability and physicochemical changes in viral droplets. These physicochemical changes in viral droplets include water activity, surface inactivation, salt toxicity [31], and phase separation of carrier aerosol constituents [32]. In evaporating viral droplets with higher organic content (proteins), solute concentration increases more slowly as compared to that of droplets containing more inorganic content at similar RH [29]. The slow increase in solute concentration represents the longer time needed to elevate their concentrations to those levels which can cause the inactivation of viruses inside aerosol particles and droplets [29]. Therefore, it may support the prolonged survival of viruses in droplets and aerosol particles with more organic content by making a coating or thick gel-type shell on top of aerosols [3,29]. In dry indoor settings, small viral droplets undergo quick evaporation and shrink to smaller sizes known as aerosol particles and as a result quickly disperse in indoor settings [11]. In the case of poorly ventilated indoor environments, the viral aerosol particles will remain in an airborne state for longer durations. Therefore, higher concentrations of organic content in droplets and aerosol particles under poorly ventilated indoor conditions could influence the indoor transmission of COVID-19 by increasing the virus lifetime inside the aerosol particles [33]. NaCl (inorganic content in droplets) because of its hygroscopic behavior can play an important role in controlling the water uptake and loss in aerosol particles. Therefore, the solute concentration increases more rapidly in droplets and aerosol particles with more inorganic content (NaCl) as compared to those with more organic content (i.e., proteins) at similar RH [29].

### 2.2. Effect of Change in pH (Due to Change in RH) on Viability of Enveloped Viruses

Another important property, i.e., pH, can also affect virus viability inside the droplets and aerosol particles. There is a change in pH values when the expelled respiratory droplets and aerosol particles either grow by water uptake or diminish by evaporation of water. This may result in additional phase transitions. Previous studies have reported that liquid-liquid phase separation and other phase transitions are strongly impacted by the pH values (Figure 2) [34]. Droplets can vary in their pH which can change with RH, and this will affect the viability of aerosolized viruses by changing the conformation of viral proteins or the electrostatic properties [20]. Here, we will first describe the effect of changes in conformations of viral glycoproteins, and in the later part, the effect of electrostatic properties such as isoelectric points related to changes in pH.

#### 2.2.1. Effect of Conformational Changes in Viral Glycoproteins Related to Changes in pH

As stated above, during the winter season, indoor RH falls below 30% due to the excessive usage of heating to maintain comfortable room air temperatures. This happens mostly when external air temperatures are low (<4 °C) with dry air (vapor pressure < 0.75 kPa) [15]. Due to low indoor RH levels, droplets will evaporate at a faster rate. This would increase the free H^+^ ions concentrations in an aerosol particle and, in turn, decrease the pH values. This results in re-arrangements in embedded glycoprotein’s structure (depends on their response to low pH values) which are present in enveloped viruses’ membranes. These glycoproteins are essential to viral attachment and subsequently for entry to host cells [31]. The exact changes in pH values after evaporation at a certain RH are far more complicated than the simple assumptions considered in previous studies [31]. There can be many possible reasons for the complication, (i) due to interaction between different solutes, (ii) buffering effects of proteins, (iii) influence of the chemical composition of the surrounding environment, and (iv) heterogeneity in the spatial distribution of different solutes in an aerosol particle [31]. This important study showed that droplets shrank resulting in an increment in the H^+^ ion concentration and consequently, a decrement in pH values [31]. In addition to a reduction in droplet size and pH due to an increase in H^+^ ions, these H^+^ tend to accumulate on aerosol particles and droplet surfaces with a partition coefficient of 1.5 (ratio of surface concentration to bulk concentration). This phenomenon would further reduce the pH values on the surface, where the accumulation of enveloped viruses takes place by additional 0.2 units as compared to within aerosol particles and droplets [31]. The changes in the pH values can be crucial for virus inactivation/survival inside the viral droplets as well as when interacting with the host cells. Before proceeding further on the role of pH in the fusion process, we need to get an idea of how the virus replicates and enters host cells. The replication process starts with fusion because of how the enveloped viruses move into the host cells. Then the fusion process is activated by the low pH of endosomes by enabling irreversible conformational changes in the glycoprotein [35]. There are two mechanisms linked to such activity (1) direct fusion with the plasma membrane or (2) fusion following endocytosis and intra-cellular trafficking (some viruses can take either pathway) [31]. From the literature, it has been found that at neutral pH values, the direct fusion process takes place for enveloped viruses, such as RSV and alpha-herpesvirus [36,37]. Our interest lies more in the enveloped viruses which require lower pH values for the fusion process. The other viruses such as IAV and Langat enter the host cells through the endocytosis pathway. To activate the fusion process, these viruses typically require low pH (5 to 6 or lower) [38,39,40]. The attachment of IAV to sialic acid is typically found which contains receptors via the hemagglutinin (HA) glycoprotein. After this process, it fuses through endocytosis. At a pH value ~5, the HA glycoprotein experiences an acid-catalyzed conformational rearrangement within the endosome. This exposes the fusion peptide, and it will subsequently fuse with the endosomal membrane [31]. The SARS viruses also trigger the fusion process at lower values of pH [41].

There can be similar acid-catalyzed conformational changes in the viral glycoproteins at low pH values. Similarly, there will be conformational changes in the glycoproteins when the acidification occurs without the target membrane existence outside the host cell. This would inactivate the virus’s fusion activity, and hence, infectivity [37]. 

#### 2.2.2. Impact of RH and pH on Virus Viability

The enveloped viruses which require acidification before fusion (such as influenza virus and SARS-associated coronavirus) were found to be less stable at intermediate RH (~ 50 to 90%) compared to higher and lower RH [20]. At 50–90% RH, the viral aerosol particles do not necessarily dry out, but the evaporation becomes more intense at around 50% RH value within the 50–90% RH range [29,37].

**Figure 2 viruses-14-01497-f002:**
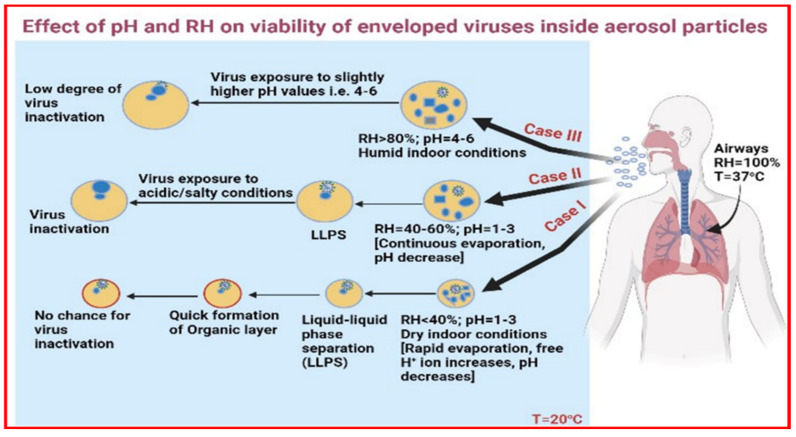
Impact of pH and RH on enveloped viruses’ viability inside viral aerosol particles (modified version [17,42].

In Figure 2, there are three different cases considered, Case I signifies dry indoor conditions due to heating indoors at T_in_ = 20 °C, RH_in_ < 40%, and outdoor temperature, i.e., T_out_ < 4 °C. Here, expelled respiratory droplets will follow rapid evaporation, and in consequence, pH decreases due to high H^+^ ions. Later, liquid-liquid phase separation (LLPS) (e.g., amorphous organic shell and aqueous inorganic core) happens at low RH and because of extremely fast evaporation, the particle dries out completely. Thus, complete dryness could in fact protect the enveloped viruses from the high concentrations of dissolved salts and low pH [14]. This means no chance for inactivation of enveloped viruses despite the acidic environment. The quick formation of the organic layer on the top ensures no inactivation takes place. Typically, a sudden increase in salt concentrations and/or the exposure time raising salt concentrations and reducing pH, likely causes damage to the virus [13,14,29]. However, pathogens that are encapsulated in a gel-type or glassy structure may be protected from oxidative damage or solute effects [3].

In Case II (T_in_ = 20 °C, RH_in_ = 40–60%), continuous evaporation takes place and pH decreases continuously providing enough time to enhance the solute concentration (with acidic conditions) in order to inactivate the virus (increased osmotic stress), this happens at an intermediate RH range ~ 40–60% [29].

Case III (T_in_ = 20 °C, RH_in_ > 80%) shows slow evaporation in more humid conditions, i.e., >80% RH with slightly higher pH values around 6 or more. At high RH, virions will be exposed because of the liquid-like state of aerosol particles and droplets, although low levels of solute concentrations are observed, and the disinfection process may continue at a slower rate [3]. This will lead to a low degree of virus inactivation [29].

As explained above, evaporation occurs at the lower indoor RH, declining the pH values, and in turn, most of the surface proteins get conformational changes. The viral glycoproteins get conformational changes below a threshold pH value (e.g., pH 5 for IAV). Further, at a specific RH, even a small amount of decrement in pH value near to threshold may activate the denaturing of glycoproteins which is an irreversible process [38]. Another important study suggests that low pH values may play an important role in the stability of enveloped viruses [43]. However, this analysis also indicates at a specific RH, the change in pH values linked with droplet evaporation may be a possible intermediary of virus viability [31].

A recent experiment conducted by Lin et al. [13] suggests that the relative viability of phi6 differed significantly with varying pH values at three different RH levels. At a pH value of 4.0, there was no viable phi6 detected in either evaporating droplets or in the control solution after 1 h of the experiment. This suggests a strong inactivation effect of acidic conditions on phi6. At a pH value ~10.0 after 1 h, the virus decayed by ~1–3 log_10_ units (RH dependent). Virus survival was highest in pH-neutral droplets (7.0) after 1 h, where it decayed just by ~1–2 log_10_ units (RH dependent). The decay of phi6 was found more evident in acidic or basic droplets. The survival of phi6 was found to be best in pH-neutral droplets. A recent study has reported that the pH effect is more pronounced on the viability of enveloped viruses than non-enveloped viruses [13]. Therefore, we have considered only enveloped viruses in our study (Table 1). It is noteworthy to mention that SARS-CoV-2 falls into the enveloped virus category, so it is extremely important to study the effect of pH on the survival of SARS-CoV-2 in aerosol particles and droplets.

#### 2.2.3. Virus Isoelectric Points and Their Dependence on pH

Here, we discuss virus isoelectric points and their dependence on pH. The pH value at which the net surface charge changes its sign is known as the isoelectric point. This is known as a characteristic parameter of the virus particles which are in equilibrium with their environmental water chemistry [51]. In the case of enveloped viruses, the surface charge of the capsid mainly controls its interaction with the lipid bilayer [52]. The capsid is the protective layer of the virus genome whereas the envelope is a protective covering of the protein capsid. However, the role of envelope phospholipids is substantial to the surface charge of the envelope. This can typically lead to a probable decrease in the apparent pKa value (strength of an acid) by almost one pH unit due to the low dielectric constant of phospholipid bilayers [53]. At pH values below the isoelectric point, there could be a positive charge on a virus, on condition when both carboxylate and amine groups on the outer surface are protonated and hydrogen bonding would be formed to hydroxyl-containing surfaces. At pH values above the isoelectric point, due to the deprotonation process on the outer surface of virions, there will be a negative charge on the virus [54]. Based on a process related to the virus’s isoelectric points, virus inactivation can be protected by favoring pH values [29]. In addition, the pH gradients present in aerosol particles and droplets would further perplex the choate process, most prominently when there is inadequate knowledge of the spatial distribution of viruses in aerosols and droplets [29]. 

Lastly, pH and RH together have a serious impact on the viability of enveloped viruses. In addition, the inactivation process may also get affected by the physical behavior of viruses, such as partitioning of the air-liquid interface and forming aggregates which result from changes in droplet characteristics. Until today, the information about the role of pH (both acidic and basic behavior) in SARS-CoV-2 virus viability is largely unknown and needs to be explored in the future.

## 3. Instrumentation Required for Measuring Expelled Viral Droplets and Aerosol Particle’s Chemical Properties

The direct measurement of aerosol pH in droplets and aerosol particles is extremely challenging yet it can be estimated with thermodynamic models based on the known chemical composition and the equilibrium among acids and their conjugate bases [55,56]. In a nutshell, there are numerous potential spectroscopic and microscopic techniques, including electron microscopy, X-ray microscopy, fluorescence microscopy, single-particle aerosol mass spectroscopy, and Raman microscopy that can be used for single-particle characterization. These can also be used for exploring the chemical properties of viral human respiratory droplets and aerosol particles [55]. The time-of-flight secondary ion mass spectrometry (TOF-SIMS) can also be used to analyze respiratory exhaled particles and for identification of phospholipids features majorly in the pulmonary surfactant [57]. Among these phospholipids, DPPC was considered the main constituent of pulmonary surfactants. Recent studies have confirmed the presence of these pulmonary surfactants through a combination of various processes such as sodium dodecyl sulfate poly-acrylamide gel electrophoresis (SDS-PAGE) [58], liquid chromatography-mass spectrometry (LC-MS), and triple quadrupole mass spectrometry (TQMS). Apart from this, Gesundheit II (G-II) which was manufactured to detect the Influenza virus in exhaled breath can be used to study the properties of viral aerosol particles with SARS-CoV-2 inside the exhaled breath [59]. Recently, a European network has initiated examining viral particles’ chemical composition for the identification of COVID-19 disease using mass spectrometers in real time within seconds [60]. This is a step forward in exploiting the existing mass spectrometers in the virology field.

## 4. Conclusions

We have highlighted the impact of chemical properties of human respiratory droplets and aerosol particles such as chemical composition and pH on airborne viruses’ viability in indoor environments which is critical to understanding the indoor viral airborne transmission process and taking necessary measures to control the ongoing pandemic. Despite several studies dealing with the impact of meteorological parameters and size on SARS-CoV-2 indoor transmission, we found limited studies on the impact of aerosol particle chemical properties on virus viability and indoor transmission. Based on the limited existing knowledge, our analysis demonstrated that the chemical properties of viral droplets and aerosol particles play a crucial role in governing the transmission of the virus. The most important findings are:High organic content in viral droplets and aerosol particles supports prolonged survival of the virus by forming a glassy structure that restricts the virus inactivation process mostly in dry indoor conditions [3].The enveloped bacteriophage Phi6 decayed more at lower and higher pH values while staying viable at a neutral pH value of 7 [13].

Further, experimental and modeling research studies on this topic are needed to confirm our hypothesis and propositions which we have provided. Furthermore, we suggest recommendations to prevent SARS-CoV-2 airborne transmission indoors, especially in poorly ventilated places such as certain hospitals, schools, and public buildings [61]. Ultimately, to fight against COVID-19, the understanding of physicochemical properties of viral droplets and aerosol particles, the antiviral drug distribution among communities at a faster scale, and the incorporation of all the measures presented in Ahlawat et al. [61] will eventually mitigate this global threat.

## Figures and Tables

**Figure 1 viruses-14-01497-f001:**
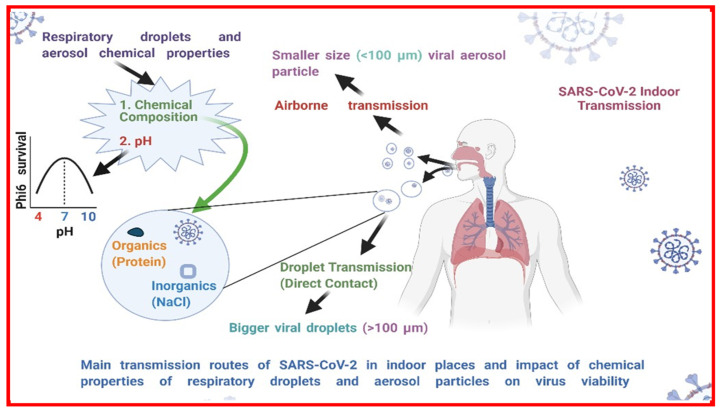
The impact of chemical properties of viral aerosol particles and droplets on the indoor transmission of SARS-CoV-2. The main transmission routes of SARS-CoV-2 in indoor environments are either due to droplets (direct contact) or aerosol particles (airborne transmission). The chemical composition and pH are important chemical properties that impact SARS-CoV-2 viability in expelled droplets. The chemical composition typically represents proteins, and NaCl along with the virus in a viral expelled fluid. Here, from the literature, we also report how pH values impact virus survival, Phi6 bacteriophage tested at different values of pH [13].

**Table 1 viruses-14-01497-t001:** Information of fusion process for various enveloped viruses, modified version [31].

Name of Virus	Fusion pH	Spray Medium	Virus Viability (References)
Influenza (PR8)	Low pH (<5)	1-part allantoic fluid plus 1 part 2% peptone	Viability highest at 15–20% RH and lowest at 40–90% RH [44,45]
Influenza A (PR8)	Low pH (<5)	Allantoic fluid diluted 1:8 or 1:10 in casein McIlvaine’s buffer (pH 7.2)	Viability decreases with increasing RH [46]
Influenza A (W.S. Strain)	Low pH (<5)	Allantoic fluid in 0.1 M Sorensen’s phosphate buffer (pH 7.1)	Viability highest at 30–34% RH, lowest at 58–60% RH and medium at 66–70% RH [47]
Influenza A (WSN Strain)	Low pH (<5)	MEM; MEM plus 0.1% BSA; allantoic fluid	Viability highest at <45% RH, lowest at 40–60% RH and medium at RH > 80% [48]
SARS-CoV-1	Low pH	Cell culture maintenance	Stable at 40% RH and more rapidly inactivated at higher RH in 1 h [49] Inactivation effect of acidic conditions (pH < 3) and alkanine (pH > 12) [50]
Phi 6	Low pH	Cell culture maintenance	Inactivation at intermediate RH, Survival at low RH; Strong inactivation effect of acidic conditions (pH < 4) and alkaline conditions (pH > 10) [13]
Langat	Low pH (<6)	Culture medium (salts and protein)	Viability highest at 20% RH, lowest at 40–60% RH, and medium at >70% RH [18]

## Data Availability

The original contributions presented in the study are included in the article, further inquiries can be directed to the corresponding author.

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
