# Peer review of "Impact of Chemical Properties of Human Respiratory Droplets and Aerosol Particles on Airborne Viruses’ Viability and Indoor Transmission"

_viruses, 2022, doi:10.3390/v14071497_

Round 1

Reviewer 1 Report

This paper is a short review of the effects of aerosol chemical properties on the survival of viruses. The review is interesting and informative. I do have a few comments that should be addressed.

 First, the title of the paper is somewhat misleading. As the authors themselves note, very little information is available on the survival of SARS-CoV-2 under different conditions. Thus, what the authors actually discuss is information about the survival of other enveloped viruses such as influenza and the bacteriophage Phi 6. The authors propose phi 6 as a surrogate for SARS-CoV-2. Unfortunately, the term “surrogate” gets used quite loosely for viruses in in the aerosols field without any explanation or justification as to why something is a good surrogate or in what sense it is a surrogate. SARS-CoV-2 and Phi 6 are both enveloped viruses, but there is no evidence that suggests that their survival rates are the same under any conditions or that they are affected in the same way by factors like pH, humidity, or salt concentration. The discussion is useful, but it is an overreach to assume that factors like pH will affect SARS-CoV-2 in the same way without any supporting evidence, particularly since the effects of these factors are not well understood. A better title would replace “SARS-CoV-2” with something like “airborne viruses”.

 The paper has a good review of the existing literature. The authors should also consider two recent papers studying the effects of mucus on influenza survival in aerosols and droplets, since they discuss this topic in some detail:

 Kormuth, KA, K Lin, AJ Prussin, 2nd, EP Vejerano, AJ Tiwari, SS Cox, MM Myerburg, SS Lakdawala and LC Marr (2018). Influenza Virus Infectivity Is Retained in Aerosols and Droplets Independent of Relative Humidity. J Infect Dis 218(5): 739-747. https://doi.org/10.1093/infdis/jiy221

 Kormuth, KA, K Lin, Z Qian, MM Myerburg, LC Marr and SS Lakdawala (2019). Environmental Persistence of Influenza Viruses Is Dependent upon Virus Type and Host Origin. mSphere 4(4). https://doi.org/10.1128/mSphere.00552-19

 Finally, the paper would greatly benefit from a review by a writer/editor. The paper has numerous sentences that are incomplete or badly structured, incorrect word choices, and awkward phrasing. In places, the paper jumps back and forth between topics, making it difficult to follow.

Reviewer 2 Report

In this study, the authors raise the very important topic of how rapidly changing chemistry in respiratory droplets and aerosol particles affects the viability and transmission of embedded viruses. Given the airborne route of spread of many dangerous viral diseases, including COVID-19, this study is undoubtedly important and relevant. However, I made a number of comments, and, after corrections, the article can be considered for acceptance into the journal Viruses.

11.       Since the authors mention the results of other studies in the abstract, it is necessary to insert references.

22.       L. 96 “Also, the viral droplet composition 96 (chemical composition, pH) changes for different types of diseases [24]” Are there data on changes in droplet size and composition during the course of the disease?

33.       It is necessary to check whether there are permissions to use fragments of figures from the articles cited in the captions.

44.       L.143 – “confirmations

55.       L.164 – “The changes in the pH values can be crucial for virus inactivation/survival inside the viral droplets as well as when interacting with the host cells.” Is it possible that the amount of respiratory droplets that enters the airway mucosa during infection can change the acidity on the surface of the mucosa and on the host cells?

66.       “Table 1. Information of fusion process for various enveloped viruses…” The last column concerns viability, so it must be indicated in the table title.

77.       Probably it would be helpful if the authors summarized the effects of pH on virus viability levels in a separate table or more specifically indicate in Table 1 what change in virus titers may occur as a function of pH.

88.       L.243…” In the case of enveloped viruses, surface charge of the capsid mainly controls its interaction with the lipid bilayer [49]. In addition, the role of envelope phospholipids is substantial to the surface charge. This can typically lead to probable decrease in the apparent pKa value (strength of an acid) by almost one pH unit due to the low dielectric constant of phospholipid bilayers [50]” Please separate the charge on the capsid and the charge on the surface of the virus envelope. Discuss in more detail what structures, molecules, groups contribute to the latter, how their functional activity can change depending on pH.
